# Cognitive Decline Secondary to Therapeutic Brain Radiation—Similarities and Differences to Traumatic Brain Injury

**DOI:** 10.3390/brainsci9050097

**Published:** 2019-04-27

**Authors:** Andrew Jonathan Huang, David Kornguth, Steven Kornguth

**Affiliations:** 1University of Wisconsin Hospital, Madison, WI 53726; USA; A.J.Huang.MD@gmail.com; 2Golden Gate Cancer Center, San Francisco, CA 94107, USA; dkornguth@gmail.com; 3Dell Medical School, The University of Texas Austin, Austin, TX 78701, USA

**Keywords:** radiation induced brain injury, traumatic brain injury, cognitive loss, inflammatory response, immune factors, receptor changes, blood brain barrier

## Abstract

Traumatic brain injury (TBI) resulting from forceful impacts on the torso and head has been of major interest because of the prevalence of such injuries in military personnel, contact sports and the elderly. Cognitive and behavioral changes associated with TBI are also seen following whole brain radiation treatment for cancer and chemotherapy for disseminated tumors. The biological mechanisms involved in the initiation of TBI from impact, radiation, and chemotherapy to loss of cognitive function have several shared characteristics including increases in blood brain barrier permeability, blood vessel density, increases in inflammatory and autoimmune responses, alterations in NMDA and glutamate receptor levels and release of proteins normally sequestered in the brain into the blood and spinal fluid. The development of therapeutic agents that mitigate the loss of cognition and development of behavioral disorders in patients experiencing radiation-induced injury may provide benefit to those with TBI when similar processes are involved on a cellular or molecular level. Increased collaborative efforts between the radiation oncology and the neurology and psychiatry communities may be of major benefit for the management of brain injury from varied environmental insults.

## 1. Introduction

Radiation treatment to the brain (for management of neoplasms) and non-penetrating traumatic brain injury (TBI) are clinical conditions that result in major changes in neural function. While the majority of traumatic brain injuries (TBI’s) are associated with falls, and vehicular accidents, this paper will have a focus on TBIs resulting from the forceful impacts of sports or military activities [1]. Decrements in cognition, mood, sensory and motor function occur over prolonged periods of time following either radiation treatment or TBI [2,3,4,5,6,7,8]. With more than 50,000 new cases of brain neoplasms each year in the United States, and patient survival reaching years instead of months, radiation induced cognitive injury is becoming an increasingly important topic of conversation [9,10,11,12,13,14]. In contrast to tumors in the brain, TBI is relatively ubiquitous with approximately 10 million events annually [15]. Only a small subset of these TBI patients die from their injuries, with only ~53,000 patients dying per year from TBI in the United States [16]. Because of the commonalities of (1) delayed decrement in neural function between the initial insult of radiation or TBI, and (2) the change in cognition and mood, we initiated this review to explore whether there are common underlying mechanisms operating in both conditions that may lead to the development of new research and therapeutic options for each condition [17,18].

In the field of neuro-oncology, radiation therapy is an effective therapeutic treatment for neoplasms across ages and cancer subtypes with the consequence of cognitive decline dependent on the age of the patient, the volume of brain irradiated and the dose delivered [19,20,21,22,23,24]. Traditionally, oncologists utilized radiation therapy rather than systemic drugs to eradicate microscopic disease in the central nervous system (CNS) because the blood-brain barrier blocks the large molecules used in classic chemotherapies [25,26,27]. While this rationale may be somewhat outdated, only a few agents have emerged that are effective in controlling central nervous system (CNS) lesions and initial results continue to support radiotherapy as an initial modality of treatment [28,29,30,31,32].

While cognitive decline secondary to radiation therapy is a problem that continues to be studied, understanding and solving this problem is difficult for several reasons: (1) wide variations in age, treatment technique, tumor types, and cognitive performance scoring make generalizations difficult, (2) most patients are not cured and thus competing sources of cognitive decline such as tumor progression and further chemotherapy confound most analyses, and (3) radiation oncologists are increasingly utilizing the technique of stereotactic radiosurgery (SRS) over whole brain radiotherapy (WBRT) which minimizes dose to the healthy brain, thus limiting the pool of patients that would be best for studying cognitive decline [33,34,35,36,37,38,39,40,41,42].

Radiation to the brain is administered in one of two treatment paradigms: (1) 5–30 treatments, each delivering a dose of 1.5–4 Gy, to a relatively wide area that encompasses normal brain, or (2) 1–5 treatments, each delivering a dose of 5–24 Gy, with minimal normal brain contained within the high dose field [43,44,45,46]. When the first paradigm is utilized and the entire brain is encompassed in the radiotherapy field, we use the term WBRT but when only a portion of the brain is targeted, we use the terms 3D conformal or intensity modulated radiotherapy. When the second paradigm is used and only a single treatment is given, we use the term SRS but when multiple treatments are given, we use the term stereotactic radiotherapy (SRT). This review will primarily focus on the adverse effects of WBRT because its effects on cognition are most pronounced and our basic science understanding of how the brain reacts to radiation is based on animal models utilizing this technique [47,48,49].

## 2. Clinical Effects

### 2.1. Cognitive Function

WBRT has global effects on the brain with measurable deficits in energy, episodic verbal learning, verbal fluency, processing speed, visual attention, task switching, and memory [2,3]. To estimate the probability of suffering these adverse effects, two trials that randomized patients with brain metastases to +/− WBRT will be considered.

Kondziolka et al. measured the quality of life of patients who had 2–4 brain metastases and received SRS +/− WBRT (30 Gy divided into 12 daily treatments) [50]. They found that in the arm that received SRS + WBRT, 99% reported hair loss, 95% reported excess fatigue, 83% had problems with short term memory, 72% had problems with concentration, 61% had problems with depression, and 40% had problems with long term memory [50]. The MD Anderson group ran a similar trial that randomized patients with 1–3 brain metastases to SRS +/− WBRT (30 Gy divided into 12 daily treatments) [51]. At 4 months, 52% of the SRS + WBRT group was estimated to have deficits in total recall, 22% for delayed recall, and 11% for delayed recognition.

Similar to patients who receive WBRT, patients who experience TBI report deficits in memory, attention, awareness, judgement, language, visuospatial processing, and general intelligence [4,5,6,7,8,52]. A portion of patients with TBI progress to an end stage disease diagnosed post mortem as Chronic Traumatic Encephalopathy (CTE). Patients with CTE progress through increasingly severe stages that are based on neuropathological factors and experience both motor and behavioral changes such as parkinsonism, aggression, depression, delusions, and impulsivity [17,53].

The localization of these deficits in TBI patients can help radiation oncologists predict the expected side effects from radiotherapy and provide potential areas to shield from radiation. For example, Spikman et al. found that patients who have imaging-confirmed frontal lobe changes frequently have deficits in self-awareness and have impaired self-evaluative ability [8]. Newcombe et al. found that patients with impaired reasoning and judgement abilities have changes to the frontal lobe as well as the basal ganglia [7]. As the brain continues to be mapped, radiation oncologists can utilize this data to set new avoidance structures and spill radiation dose into less critical regions of the brain.

### 2.2. Time Course

In patients who have multiple brain lesions and receive WBRT, their cognitive function declines steadily over the course of months to years [2,3,54,55]. Because WBRT is most often palliative in intent rather than curative, much of this cognitive decline is a result of the regrowth of the brain metastases themselves, new brain metastases, and subsequent systemic therapies.

Our group’s hypothesis regarding the effect of WBRT on brain injury is that there are two processes, acute and delayed, that have differing time courses and cognitive effects. When the entirety of a healthy adult brain is exposed to radiation, the patient will suffer acute declines in processing speed and alertness within 1–6 months that are the result of direct damage on neuronal stem cells. However, in the absence of tumor progression, a proportion of these patients will recover from these acute effects. At the same time, a subset of patients will experience delayed effects from radiotherapy that result from the subtler changes in the brain that result from receptor level changes and chronic neuroinflammation. Patients that receive higher radiation doses to the whole brain appear far more likely to suffer from these delayed cognitive deficits.

Somewhat similarly to the acute effects of WBRT, impairment from TBI is worst at the time of the incident and patients typically improve over months before reaching a plateau within a year [56]. In patients who suffer mild TBI, 67%–92% of patients have symptoms that resolve by 3 months time [57]. Like the chronic effects of WBRT, the cognitive decline of individuals with persistent TBI progresses over time but that is where the similarities end. Persistent TBI manifests after a dormant period, typically 10–30 years after the start of brain injury [58]. The progression of TBI usually occurs over a period of decades [59]. This progression is also much more severe than patients who undergo WBRT - in patients who donated their brains to the Center for the Study of Traumatic Encephalopathy, the majority who were assessed to have CTE ultimately died from suicide, starvation or respiratory failure [59,60].

While the time courses of radiation-induced and trauma-induced cognitive decline are not congruent, it does give us clues on the speed at which the brain heals itself and on the cumulative effects of chronic inflammation.

### 2.3. Sites of Injury

#### 2.3.1. Differential Sensitivity of Anatomic Regions

Trauma and radiation therapy both preferentially damage certain regions of the brain. Radiation appears to have preferential effects on the para-hippocampal cingulum, cuneate cortex, prefrontal cortex, cerebellum, and thalamus [61]. Clinicians have primarily focused their efforts at minimizing radiation dose to the hippocampus due to its role in memory [62]. When the hippocampus is exposed to radiation, neural stem cells in the hippocampus become less proliferative, more apoptotic, and more likely differentiate into glial cells rather than neuronal cells [63,64,65]. When adult rats are exposed to fractionated brain radiation, their ability to form new neurons in the dentate gyrus of the hippocampus is severely impaired, which results in poorer short-term memory at 8 and 21 days post-radiation [66]. These deleterious effects are potentially mediated by the resultant inflammation rather than direct DNA damage by radiation and a pre-clinical mouse model has demonstrated that suppressing the recruitment of myeloid precursors to the site of injury (by inhibition of the Colony Stimulating Factor-1 receptor) prevents cognitive injury after radiotherapy [63,67].

Initial human trials have focused on decreasing the dose to the hippocampus as much as possible rather than squelching the inflammatory effects. Gondi et al. demonstrated in their phase II trial that keeping the hippocampal dose below 9 Gy (given over 10 treatments) resulted in less decline in the Hopkins Verbal Learning Test (HVLT) versus historical controls (7% versus 30% at 4 months) [68]. The ongoing NRG-CC003 trial which compares prophylactic cranial irradiation with or without hippocampal sparing will confirm these findings in a randomized fashion [69].

In the case of TBI, the primary sites of injury are traditionally characterized as being greatest at the base of the sulci near the site of impact (the coup) and the area of the brain opposite this region (the contre-coup) [70]. More recently, MRI neuroimaging has allowed us to detect more subtle sites of injury. We understand now that the frontal and temporal regions of the brain are frequently injured because the concavities in the skull base cup these regions and the sulci of the brain often widen, because of the water hammer effect [71,72,73]. The water hammer theory of TBI states that the pressure surges caused by the sudden change in motion of the cerebrospinal fluid (CSF) is sufficient to damage and eventually widen the sulci of the brain. The deficits seen in the executive functions of the brain and memory processing support the sensitivities of the frontal and temporal regions respectively [4,74]. In a manner similar to the effort to protect sensitive brain regions from injury following radiation therapy, helmet makers can focus efforts on decreasing accelerative forces in vectors that are most likely to cause TBI.

#### 2.3.2. Blood Vessels

Both trauma and ionizing radiation temporarily disrupt the permeability of the blood-brain-barrier [49,75,76,77]. This disruption of the blood brain barrier is one of the cornerstones for the autoimmune theory of progression of persistent TBI to CTE which revolves around the concept that neural proteins are released from the brain after trauma [59,73,77]. A subset of TBI patients will develop antibodies to these neural proteins that then result in injury to neural cell populations, neuroinflammation and end stage CTE.

Radiation induces increased levels of neuronal biomarkers in the CSF and bloodstream like neurofilament light protein (NFL), and T-tau but NFL release is thought to be independent of blood-brain barrier permeability [78,79]. Work is currently being done to develop a signature profile of persistent TBI that measures the levels of the aforementioned biomarkers as well as other glial or neuronal markers like GFAP, NF heavy, amyloid beta, and MAP 2; this assay may prove to be useful in assessing the degree of injury from radiation as well [80].

In the long term, radiation appears to reduce both blood vessel density and blood vessel length in the brain which likely plays a large role in the long-term detrimental effects of radiotherapy. Brown et al. irradiated young-adult rats with fractionated WBRT (40 Gy in 8 treatments over 5 weeks) and dissected their brains at several time points over a year [81]. They found that between irradiated and non-irradiated rats, vessel density and length were the same 24 h after the last radiation treatment but at 10 weeks post-treatment, the irradiated rats had substantially decreased vessel density and length compared to the control rats. Vessel density and length then gradually decreased over the course of a year at a similar rate as for the control rats.

### 2.4. Receptor Level Changes

#### 2.4.1. Changes in Neuronal Receptors

When fetal mouse neuronal stem cells are exposed to radiation in-vitro, they undergo an alternate differentiation pathway that results in neurons with a higher expression of glutamate receptors [82]. In young rats, ex vivo radiation of hippocampal cells led to early decreases in tyrosine phosphorylation and removal of excitatory N-methyl-D-aspartate (NMDA) receptors from the cell surface while simultaneously increasing the surface expression of inhibitory gamma-aminobutyric acid (GABA) receptors [83]. These alterations in cellular localization corresponded with altered synaptic responses and inhibition of long-term potentiation. In this ex vivo model, memantine blocked these radiation-induced alterations in cellular distribution but when it was tested in humans in a phase III clinical trial it was only mildly helpful in reducing cognitive decline. Fluoxetine has recently shown promise in a mouse model to restore long-term potentiation, decrease anxiety, and decrease memory loss after radiation and temozolomide [3,84].

When adult rats were exposed to cranial irradiation, their NMDA receptors on their hippocampal neurons showed additional NR1 and NR2A subtypes compared to non-irradiated controls but their AMPA receptors levels were unchanged [85]. In a separate experiment, radiation yielded transient upregulation of HOMER-1a expression in the hippocampus but down regulation in the cortex [86]. Two months later, Homer1a expression correlated with a down-regulation of the hippocampal glutamate receptor 1 and protein kinase C γ, and an up-regulation of cortical glutamate receptor 1 and protein kinase C γ. Drugs that affect angiotensin, L-158,809, and ramipril, reversed these receptor changes. Ramipril has since been tested in other rodent models showing protective effects in various organs but no group to our knowledge has tested the radioprotective effects of ramipril in humans [87,88,89].

Receptor level changes in cannabinoid, benzodiazepine, aquaporin 4, vimentin, NMDA, AMPA, and GABA receptors have been recorded after TBI in rodent models [90,91,92,93]. Targeting these receptors have yielded several promising therapies in pre-clinical models. In rats who experienced TBI, scavenging blood glutamate with either oxaloacetate or pyruvate reduced the loss of neurons in the hippocampus [94]. When the pro-apoptotic protein “apoptosis-inducing factor” was down regulated in mice, glutamate induced apoptosis was reduced by 37% [95]. In an embryonic mouse model, statin drugs made cortical neurons more resistant to NMDA-induced excitotoxic death [96]. While the majority of neuroprotective agents tested in phase III clinical trials have yielded null results, new therapeutic targets continue to be trialed in the context of greater understanding of the basic science pathways that result from TBI [97,98,99].

#### 2.4.2. Neuroinflammation

Both radiation and trauma cause neuroinflammation in both the acute and chronic settings. In the setting of radiation, the acute neuroinflammation seen after radiation is more immediate and likely attributable to proinflammatory cytokines directly secreted by damaged neurons and microglia [100].

After C57BL/6 mice were irradiated with a single-dose of cranial irradiation between 0–35 Gy, transcript levels of inflammatory cytokines were elevated and glial and endothelial cells were activated [101]. When these mice received at least 15 Gy, a significant T-cell, MHC II+ cell, and CD11c+ cell influx was seen at the 30-day time point which persisted at 90, 180, and 365 days from irradiation. This persistent presence of immune cells was most pronounced in mice irradiated with 35 Gy. Further research from the same Rochester group found that CCR2 signaling loss resulted in decreased numbers of these infiltrating cells at 6 months specifically in cells that also expressed MHC-II molecules, giving a potential therapeutic target in humans [102].

Other drugs with therapeutic potential are the peroxisome proliferator-activated receptor (PPAR) agonists which regulate inflammatory signaling [103]. When F344 rats were given the PPARγ agonist pioglitazone before, during, and after being irradiated with 40 Gy (8 treatments over 4 weeks), they did not experience any radiation induced cognitive impairment (as measured by the object recognition test) [104]. This protective effect was seen when the rats received only 4 weeks of post-radiation pioglitazone. Because of the results of this trial, the safety of pioglitazone was tested in human subjects with both primary and metastatic brain tumors and phase II trials are currently in development for glioblastoma multiforme patients [105].

The same enthusiasm for mediating neuroinflammation exists in the TBI research community and dozens of pre-clinical experiments have shown significant reductions in TBI when anti-inflammatories are given immediately after injury [106]. One theory of TBI is that it is attributable to autoimmune processes that occur as a result of the disruption of the blood-brain barrier [107]. With this large body of evidence showing the deleterious effects of inflammation after trauma, initial enthusiasm for anti-inflammatory drugs were high but unfortunately human trials have shown mixed results. Early data showed promising results for the use of interleukin-1 antagonists, tumor necrosis factor alpha antagonists, and progesterone for reducing cognitive decline and even improving survival [108,109,110]. On the other hand, in the CRASH trial (corticosteroid randomization after significant head injury), patients with a Glasgow coma score ≤14 were randomized to methylprednisolone or placebo and the methylprednisolone group had a greater death risk regardless of injury severity [111].

Though the CRASH trial had negative results in TBI patients, glucocorticoid use in cranial radiotherapy patients has tremendous potential as glucocorticoids reduce the vasogenic edema commonly associated with larger tumors and reduce intracranial pressure [112]. Almost all patients who have symptoms from their brain tumors are prescribed dexamethasone and these patients who took dexamethasone while undergoing cranial irradiation may have unintentionally reduced the cognitive detriments of radiotherapy. This may partially explain how many patients do not experience cognitive decline despite receiving high doses of radiation to their brain.

## 3. Conclusions

There are some similarities between the symptoms, pathological processes and cellular level changes that occur after either radiation or forceful impact trauma to the brain which gives hope that therapeutic targets that are effective for one may be applied to the other. The potential role of autoimmune processes, as components of the mechanism resulting in cognitive and behavioral loss following exposure to radiation or forceful impacts, may enable the development of pharmaceutical or other treatments to mitigate these injuries.

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
