# Peer review of "Cognitive Decline Secondary to Therapeutic Brain Radiation—Similarities and Differences to Traumatic Brain Injury"

_brainsci, 2019, doi:10.3390/brainsci9050097_

Reviewer 1 Report

While the topic is good, there is little justification for the need of the paper. The introduction is lacking with the rationale for this study. There is no clear objective of the study and it is not immediately clear about the type of "study". It seems to be a review.

Regarding citations and references: 

Please put the references at the end of the sentences. The current style decreases manuscript readability.

Please make sure that the citations are accurate. For instance, on line 229-230, it is stated : One theory...that are delayed by years from the initial insult." A quick review of the article cited does not support this view. 

On line 116, please spell out CTE. Furthermore, please give a brief description of CTE. Similarly, on line 147, describe the water hammer effect.

In the conclusion section, the authors state: "There are sufficient similarities between the symptoms, pathological processes and cellular level changes that occur after either radiation...may be applied to the other." This is a bold conclusion that should be backed up by a concrete review of the literature and an explanation of the biological plausibility/rationale. Rewording this to a less bold statement would be more accurate.

Overall, this paper has merit, but the purpose and scope of the paper is not clear. A more systematic approach is needed and the goals and objectives needs to be clearly stated.

Author Response

Reviewer 1 states that the purpose and the scope of the paper is not clear. The authors concur with this objection and have markedly changed the first and fourth paragraph of the introduction (pg 1 and 2 of the revised). We now indicate that there are common features of neural, receptor, immunological  and behavioral changes that are seen in radiation induced brain injury and TBI. The authors also point out in the first paragraph the numbers of persons affected by radiation therapy compared with those suffering from TBI and the potential benefits that may be derived when the consequences of both injuries are examined in detail. By modifying the fourth paragraph of the introduction on pg 1 the authors provide focus to the paper. We are comparing the effects of whole brain irradiation with the effects of TBI. This mitigates the issue of the benefit of stereotactic radiation treatment vs whole brain radiation. The SRS issue is more complicated. 

Reviewer 1 also states that the authors should be more cautious in the conclusion. The authors concur and have modified the conclusion accordingly.

The references are now placed at the end of the sentence as recommended by the reviewer.

The authors have followed all the recommendations of this reviewer and anticipate that the manuscript will be accepted for publication.  

Reviewer 2 Report

The present manucript as a Perspective demonstrated similarlities and differences between therapeutic brain radiation and traumatic brain injury. The perspective was well-written briefly.

There were some errors in text; therefore, the authors should correct them.

Some abbreviations and the full-names were written as mixture. The full-name was needed only at first appearance in text.

Line 88-89, "Stuss and Alexander.....self-evaluative ability48", The authors of reference number 48 were not Stuss and Alexander. Their paper might be reference number 77. Correct it.

Author Response

The authors have modified the manuscript to conform to the recommendations of reviewer 2. The citations and references have been corrected in the revised manuscript. .